# Her2 activation mechanism reflects evolutionary preservation of asymmetric ectodomain dimers in the human EGFR family

Anton Arkhipov[1], Yibing Shan[1]*, Eric T Kim[1], Ron O Dror[1], David E Shaw[1,2]*

[1]D. E. Shaw Research, New York, United States; [2]Center for Computational Biology and Bioinformatics, Columbia University, New York, United States

**Abstract** The receptor tyrosine kinase Her2, an intensely pursued drug target, differs from other members of the EGFR family in that it does not bind EGF-like ligands, relying instead on heterodimerization with other (ligand-bound) EGFR-family receptors for activation. The structural basis for Her2 heterodimerization, however, remains poorly understood. The unexpected recent finding of asymmetric ectodomain dimer structures of *Drosophila* EGFR (dEGFR) suggests a possible structural basis for Her2 heterodimerization, but all available structures for dimers of human EGFR family ectodomains are symmetric. Here, we report results from long-timescale molecular dynamics simulations indicating that a single ligand is necessary and sufficient to stabilize the ectodomain interface of Her2 heterodimers, which assume an asymmetric conformation similar to that of dEGFR dimers. This structural parallelism suggests a dimerization mechanism that has been conserved in the evolution of the EGFR family from *Drosophila* to human.

*For correspondence: Yibing.
Shan@DEShawResearch.com (YS);
David.Shaw@DEShawResearch.
com (DES)

**Competing interests:** The authors declare that no competing interests exist.

**Reviewing editor**: Axel T Brunger, Howard Hughes Medical Institute, Stanford University, United States

## Introduction

Her2 (also known as Neu or ErbB2), a receptor tyrosine kinase belonging to the human epidermal growth factor receptor (EGFR) family that also includes EGFR/Her1, Her3, and Her4, is an important component of cell-signaling networks, and is implicated in the growth of a variety of cancers (*Hynes and Lane, 2005*; *Riese et al., 2007*; *Baselga and Swain, 2009*; *Lemmon and Schlessinger, 2010*). The receptors of the EGFR family activate through dimerization (*Figure 1A*), which is promoted by the binding of ligands from the EGF family (*Ushiro and Cohen, 1980*; *Schreiber et al., 1983*; *Chung et al., 2010*) to the receptors' extracellular regions ('ectodomains'). This activation process relies on a number of allosteric interactions in the extracellular, transmembrane, and intracellular portions of the receptor (*Endres et al., 2011*, *2013*; *Arkhipov et al., 2013*), which lead to the formation of a specific asymmetric active dimer of the intracellular kinase domains (*Zhang et al., 2006*). Her2 is unique in the family in that it does not homodimerize under normal conditions, and its ectodomain does not bind ligands. Instead, it activates through heterodimerization with other members of the family (EGFR and Her3, in particular) when they are bound to ligands (*Citri and Yarden, 2006*; *Baselga and Swain, 2009*).

The complexity of signaling underpinned by homo- and heterodimerization of EGFR family members emerged relatively recently in evolution. EGFR families in invertebrates have only one member, but gene duplication gave rise to two-member EGFR families in early vertebrate species, such as fishes. Further gene duplication eventually generated the four members of the mammalian EGFR family (*Stein and Staros, 2000*). The increase in the number of EGFR family members, accompanied by increasing diversity of the extracellular ligands they interact with, gave rise to a complex signaling network: homo- and heterodimerization of various EGF receptors, induced by ligand binding, began to generate a variety of unique signaling outputs controlled by the identity of the bound ligands.

**eLife digest** ErbB proteins are found in most multi-cellular organisms, and are involved in the regulation of a number of important cellular processes, including proliferation, migration, and differentiation. Humans have four ErbB proteins, which span the plasma membrane of cells. These proteins respond to interactions with molecules outside the cell—such as growth factors and hormones—by sending signals along the appropriate signaling pathway within the cell.

ErbB proteins have three portions: an ectodomain that extends outside the cell; a single helix that spans the membrane; and a cytoplasmic domain inside the cell. When a signaling ligand molecule outside the cell binds to the ectodomain of an ErbB protein, this protein must then combine with another ErbB protein to form a dimer before a signal can be sent within the cell. These dimers can include two copies of the same ErbB protein or two different ErbB proteins. However, one of the ErbB proteins—Her2—works in a different way. It cannot bind ligands outside the cell, and it can only send a signal within the cell if it first forms a dimer with an ErbB protein of another type, which itself must be bound to an external ligand.

The four ErbB proteins diverged from a common ancestor relatively recently, yet they are now diverse enough to play key roles in a variety of complex signaling networks. In particular, the fact that Her2 cannot bind external ligands, and that it must form a dimer with a different ErbB protein before it can send a signal, has led to suggestions that the role of Her2 is to amplify the signals from other ErbB proteins. Since high levels of Her2 are associated with aggressive forms of breast and ovarian cancer, understanding how it is activated could improve our understanding of these cancers.

Arkhipov et al. have now used computer simulations to model how Her2 forms dimers with other ErbB proteins in human cells. They based these simulations on crystal structures of human ErbB proteins and dEGFR, a growth-factor receptor found in fruit flies that closely resembles the ErbB proteins found in humans. They found that the dimers were stable as long as one protein within the dimer was bound to a ligand. Removing this ligand, however, distorted the ectodomain of the host protein, creating a gap that weakened the dimer and prevented Her2 from sending a signal within the cell. Similar results were obtained with the fruit fly dEGFR proteins. These simulations suggest that ErbB proteins form dimers and send signals through a mechanism conserved in evolution. Research in this field might help ongoing efforts to develop new treatments for human tumors characterized by high levels of Her2 expression.

The unique dimerization properties of Her2 are believed to be essential to its critical role as a potent signal amplifier for the other receptors of the EGFR family. The Her3–Her2 heterodimer is particularly prevalent and potent in signaling. The prominence of Her3–Her2 heterodimers is particularly intriguing, since Her2 itself lacks an activating ligand and Her3 is impaired in its kinase activity (*Figure 1A*). These receptors act primarily through heterodimerization. The unique partnership of Her2 and Her3 has been aptly dubbed 'the deaf and the dumb' (*Citri et al., 2003*). By activating several downstream pathways, including those of MAPK, PI3K, phospholipase C, protein kinase C, and Janus kinase, the Her3–Her2 heterodimer plays a pivotal role in the determination of cell lineage in a variety of tissues in epithelial organs. Experiments have shown that knocking out Her2 and Her3 genes leads to defective development of the heart, mammary gland, and nervous system in mice phenotypes. It is thus not surprising that Her3–Her2 heterodimers are implicated in a number of forms of cancer (*Citri et al., 2003*) or that Her2 is an important drug target in cancer therapeutics.

The structural mechanism underlying Her2's dimerization properties remains incompletely understood. In particular, the conformations of Her2 homo- and heterodimers remain obscure, and how these conformations relate to the dimerization affinities is uncertain. Moreover, the manner in which ligands binding to Her2's heterodimerization partners regulate the conformations and stability of the heterodimers is poorly understood. Although various structural and electrostatic factors that may affect Her2 dimerization have been discussed in the literature (*Cho et al., 2003*; *Franklin et al., 2004*; *Garrett et al., 2003*; *Alvarado et al., 2009*, *2010*; *Liu et al., 2012*), further structural elucidation is needed for a better molecular understanding of Her2 activation.

Crystal structures of the ectodomains of human EGFR-family receptors (*Lemmon, 2009*) provide a starting point for elucidation of the structural mechanisms of Her2 heterodimerization. In these

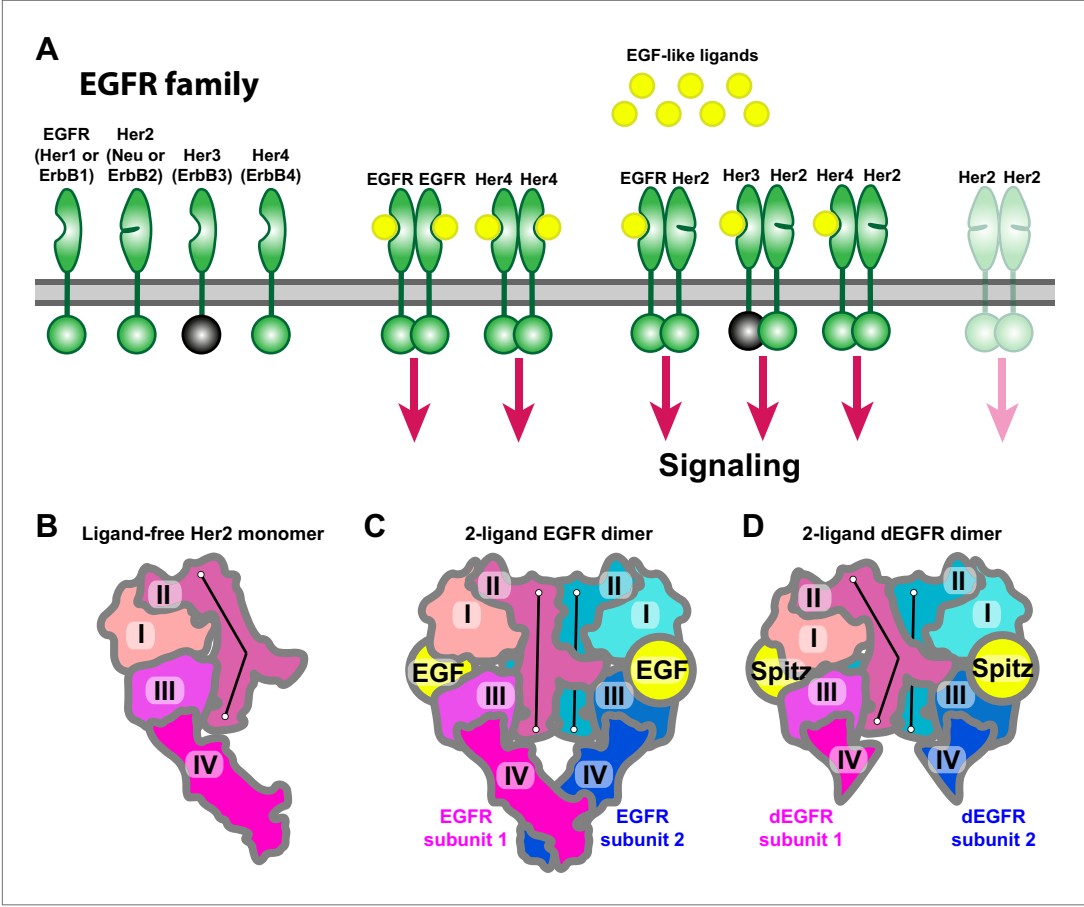

**Figure 1**. Receptors of the human EGFR family and conformations of their ectodomains. (**A**) Left: the four members of the human EGFR family, each consisting of an ectodomain, a single-pass transmembrane helix, and an intracellular module that includes a kinase domain. As shown, Her2 bears a closed ligand binding site and does not bind EGF-like ligands. Her3 is kinase-dead, and its intracellular module is thus colored gray. Right: common homo- and heterodimers of the EGFR family. The Her2 homodimer is rendered semitransparent to indicate its instability in normal cell conditions. (**B**) Schematic of the ligand-free Her2 ectodomain monomer, as observed crystallographically (PDB entries 1N8Z, 2A91, 1S78, 3N85, and 3MZW). Domain II is bent. (**C**) Schematic of the crystal structure of the 2-ligand EGFR ectodomain dimer (PDB entry 1NJP). The dimer is symmetric, and domain II is straight in both subunits. (**D**) Schematic of the crystal structure of the dEGFR ectodomain dimer (PDB entry 3LTF). Although Spitz ligands are bound to both subunits, the structure is asymmetric; domain II is straight in one subunit but bent in the other. Domain V and part of domain IV were not resolved in this crystal structure and are not shown in the schematic. The conformations of the bent and straight domain IIs in (**B**), (**C**), and (**D**) are indicated by the black lines.

structures (*Figure 1B,C*), the ectodomains (each consisting of four domains numbered I, II, III, and IV) are found either as monomers or as symmetric homodimers with ligands bound to both dimer subunits (*Garrett et al., 2002*; *Lu et al., 2010*; *Ogiso et al., 2002*; *Liu et al., 2012*). In the homodimer structures, domain II constitutes most of the ectodomain's dimer interface (*Figure 1C*), and its conformation is critical to ectodomain dimerization. A bent domain II is found in all ligand-free, inactive ectodomains of the EGFR-family receptors (*Figure 1B*), while a straight domain II is associated with the (ligand-bound) active dimer (*Figure 1C*). The domain II of Her2 appears to be constitutively bent (*Cho et al., 2003*; *Garrett et al., 2003*; *Franklin et al., 2004*; *Alvarado et al., 2009*), which is consistent with its poor homodimerization. Unfortunately, no crystal structure is available for human EGFR family heterodimers with only a single ligand bound, as is the case for Her2 heterodimers.

Intriguingly, recent crystal structures of *Drosophila* EGFR (dEGFR) ectodomains reveal asymmetric dimers that bear only one fully formed ligand binding site, with the other partially closed, regardless of whether one or both subunits are ligand-bound (*Alvarado et al., 2009*, *2010*). These dEGFR

structures hint at a possible structural mechanism for Her2 heterodimerization, but translating the dEGFR results to human Her2 is difficult due to differences between these receptors. First, dEGFR dimers assume an asymmetric conformation even when ligands are bound to both dimer subunits, whereas all existing human EGFR-family dimer structures exhibit a symmetric conformation. Second, the ectodomains of Her2 and other members of the human EGFR family each comprise four domains, whereas the dEGFR ectodomain includes a fifth domain, consisting of 166 residues immediately N-terminal to the transmembrane helix.

Here we investigate the dimerization of Her2 using molecular modeling and long-timescale molecular dynamics (MD) simulations. We modeled and simulated the EGFR–Her2 and Her3–Her2 heterodimers as well as the Her2 homodimer. The heterodimers were found to be stable when a ligand was bound to the EGFR or Her3 subunit. Our simulations further showed that, when the single bound ligand was removed from a Her2 heterodimer, a substantial gap developed in the dimer interface. A similar gap was also observed in our simulation of the Her2 homodimer, which explains the weak homodimerization of Her2. Structural analysis shows that such a gap arises from the bending of the domain IIs in both subunits due to the absence of bound ligands. These observations are strikingly similar to the findings from the crystal structures of dEGFR homodimers (*Alvarado et al., 2009*, *2010*): the ligand-free, but not the ligand-bound, dEGFR dimer exhibits a gap in the dimer interface, which explains the reduced dEGFR dimerization in the absence of ligands. In agreement with the recent experimental data showing that a single bound ligand is sufficient to activate an EGFR dimer (*Liu et al., 2012*), our simulations demonstrate that the activation mechanism of the receptors of the human EGFR family and of Her2 in particular conserves *Drosophila* EGFR's capacity to form stable asymmetric ectodomain dimers upon the binding of a single ligand.

## Results

### Simulations reproduce the gap in the interface of the dEGFR dimer

The general approach of this study was to use long-timescale MD simulations to infer from crystal structures of EGFR ectodomain dimers the structures of dimers with different constituent receptors or with different bound ligands. To examine the validity of this approach, we first applied it to dEGFR, for which the structures of both the ligand-bound and ligand-free ectodomain dimers (*Alvarado et al., 2009*, *2010*) have been resolved. Our simulations correctly demonstrated that, once the ligands are removed from the ligand-bound structure, a substantial gap between the two subunits develops, leading to a reduced dimerization interface.

Unlike the symmetric human '2-ligand' EGFR dimer (in which both subunits bear fully formed and occupied ligand binding sites; *Figure 1C*), the 2-ligand dEGFR dimer is asymmetric: although both ectodomain subunits are ligand-bound, only one bears a fully formed ligand binding site and a straight domain II (*Figure 2A*). The '1-ligand' dEGFR dimer (in which only one subunit is ligand-bound) is essentially identical to the 2-ligand dimer, except for the missing ligand. The gap between the two subunits is closed in both dEGFR dimers. Conversely, the ligand-free dEGFR dimer is symmetric, featuring a bent domain II in both subunits, and a gap in the dimer interface. We removed both ligands from the crystal structure of the 2-ligand dEGFR dimer (PDB entry 3LTF) and simulated the resulting system. In two independent simulations, a substantial gap emerged in the dimer interface, and the resulting conformation resembled that of ligand-free dEGFR crystal structures (*Figure 2A,B*). *Figure 2C* shows a significant reduction in the surface area buried within the dimer interface, consistent with the buried areas in the crystal structures of the ligand-free dEGFR dimers. We note that the dimer conformation drifted away from that of the ligand-free crystal structure after ~2 μs. This reflected the high flexibility of the simulated dimer, likely a result of the truncation of the domain IVs in the resolved crystal structure from which the simulations were initiated and the lack of inter-subunit contact between the two domains. The fact that our dEGFR simulations were able to reproduce the gap in the dimer interface found in the crystal structures of ligand-free dEGFR dimers lends support to the approach we adopted for the analysis of Her2 dimerization.

### Models of the EGFR–Her2 heterodimer and Her2 homodimer

The 2-ligand EGFR homodimer has been resolved crystallographically (*Garrett et al., 2002*; *Ogiso et al., 2002*; *Lu et al., 2010*) and studied previously using MD simulations (e.g., *Tynan et al., 2011*; *Zhang and Wriggers, 2011*). Given the close homology between the EGFR-family receptors ('Materials and methods'), this crystal structure may serve as a starting point for the modeling of Her2 homo- and heterodimers.

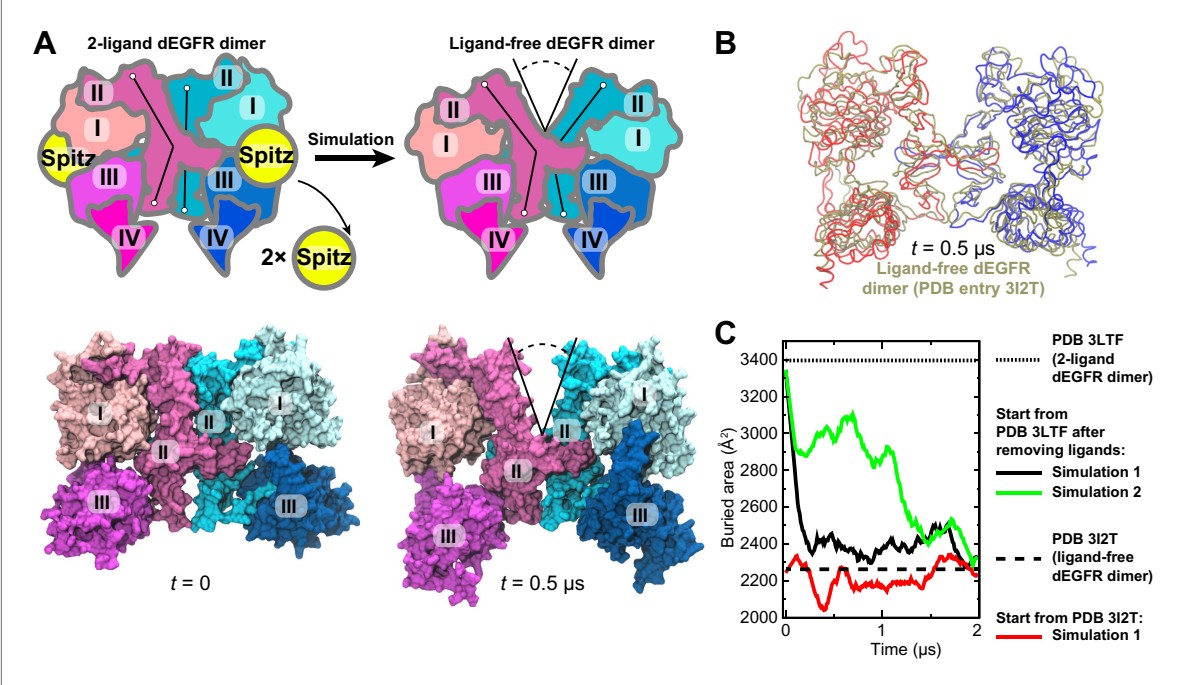

**Figure 2**. Simulations reproduce the gap in the interface of the dEGFR ectodomain dimer. (**A**) The schematic shows the 2-ligand dEGFR ectodomain dimer on the left. In simulations initiated from this structure with both ligands removed, gap opened in the dimer interface, as indicated by a V-shaped outline in the right diagram. In the ligand-free dimer, domain II is bent in both subunits. The simulation snapshots are shown below the schematic diagrams. (**B**) The simulated dEGFR dimer from (**A**), at $t = 0.5$ μs, is compared with the crystal structure of the ligand-free dEGFR dimer (tan). Molecular renderings (**A**, **B**) omit domains IV and V for clarity (although the crystallographically resolved portion of domain IV was present in simulations). (**C**) The surface area buried within the dimer interface (counting the contributions from domains I, II, and III). The results of three independent simulations are shown, two starting from the 2-ligand crystal structures (after removal of the ligands), and one from the ligand-free crystal structure.

The first model we constructed was that of the ligand-bound EGFR–Her2 heterodimer, in which the EGFR subunit is ligand-bound and the Her2 subunit is ligand-free. We assumed that a 1-ligand EGFR homodimer is structurally similar to this EGFR–Her2 heterodimer and thus may serve as a template. To obtain a structure of the 1-ligand EGFR homodimer, we removed one ligand from the crystal structure of the 2-ligand EGFR homodimer (PDB entry 3NJP [*Lu et al., 2010*]) and simulated the remaining complex (*Figure 3A*). In all three independent simulations performed, the dimer assumed an asymmetric conformation, in which the ligand-free subunit differs from the ligand-bound one. In the former, the space previously occupied by the bound ligand between domains I and III is closed and domain II is bent (*Figure 3B*), in agreement with a recent MD study (*Tynan et al., 2011*). It is notable that the removal of one of the two ligands from the ectodomain dimer did not lead to substantial decrease in the area of the dimer interfaces (*Figure 3E*), suggesting that, in agreement with recent experimental findings (*Liu et al., 2012*), a single bound ligand is sufficient to maintain a stable ectodomain dimer of EGFR. By comparison, no significant conformational changes were observed in the control simulations of the 2-ligand EGFR dimer.

The conformation of the ligand-free subunit in the simulated 1-ligand EGFR homodimer resembles the crystal structure of Her2 monomer (*Figure 3C*), especially in the bending of domain II. (The bent domain II is also observed in the crystal structures of ligand-free EGFR monomers [PDB entries 1NQL and 1YY9; *Ferguson et al., 2003*; *Li et al., 2005*].) The bending in domain II can be characterized by the angle $\theta$ formed by the top, middle, and bottom regions of the domain II dimerization interface (*Figure 3A,D*). In our simulations, this angle was ~150° to 160° in the ligand-free subunits and ~170° in the ligand-bound subunits, with the former values being very close to that of the Her2 crystal structure (156°). The EGFR–Her2 heterodimer was thus modeled by superimposing a crystal structure of the monomeric Her2 ectodomain (PDB entry 3BE1 [*Bostrom et al., 2009*]) onto the ligand-free EGFR subunit in the 1-ligand EGFR dimer using the portion of domain II directly involved in dimerization (EGFR residues 240–309; see 'Materials and methods').

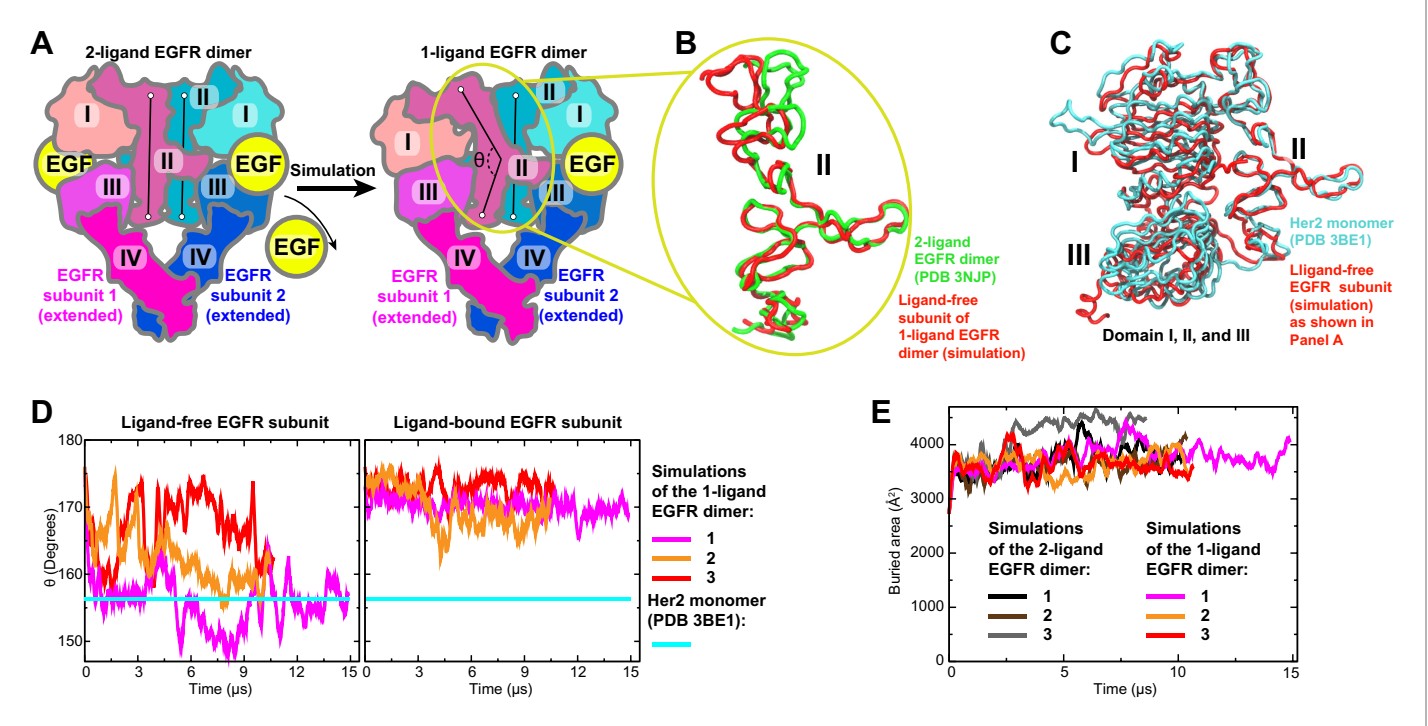

**Figure 3**. Bending of domain II. (**A**) A schematic showing how the model of the 1-ligand EGFR dimer (right) is generated by simulation after the removal of one EGF ligand from the crystal structure of the 2-ligand dimer (left). The resulting structure is asymmetric: domain II of the ligand-free subunit is bent and the binding site is closed, whereas domain II of the ligand-bound subunit is straight and the binding site is open (as is the case in both subunits of the 2-ligand dimer). The angle θ, which characterizes the bending of domain II, is measured between the Cα atoms of EGFR residues 194, 239, and 296. (**B**) Bending of domain II. A simulation snapshot of the ligand-free subunit (red) from the 1-ligand EGFR dimer is overlaid with the crystal structures of a subunit from the 2-ligand EGFR dimer (green). EGFR residues 240–309 are used for reference. (**C**) An overlay of the ligand-free subunit of the 1-ligand EGFR dimer, generated by simulation (red), with the crystal structure of Her2 monomer (cyan). In the interest of clarity, domain IV is not shown. (**D**) The angle θ (illustrated in [**A**]) is shown as function of time in the three independent simulations of the 1-ligand EGFR homodimer, for the ligand-free subunit (middle) and for the ligand-bound subunit (right). The value of θ in the crystal structure of Her2 monomer is indicated by a straight line. (**E**) The buried surface area at the interfaces of 1-ligand and 2-ligand EGFR dimers in simulation.

We subsequently modeled the ligand-free EGFR–Her2 heterodimer and the Her2 homodimer using a similar approach. In the former case, the ligand was removed from our model of the EGFR–Her2 heterodimer and the resulting ligand-free heterodimer was then simulated. In the simulations, domains I, II, and III of the EGFR subunit again assumed a 'Her2-like' conformation. This model of the ligand-free EGFR–Her2 heterodimer was then used as a template for the Her2 homodimer, in the same way that the model of the 1-ligand EGFR homodimer was used as a template for the EGFR–Her2 heterodimer ('Materials and methods').

## Dynamics of the dimer interfaces in the EGFR–Her2 heterodimer and Her2 homodimer

We subsequently simulated the EGFR–Her2 heterodimer and Her2 homodimer starting from these models. In these simulations, the ligand-bound EGFR–Her2 heterodimer remained stable (*Figure 4A*), the dimer interface was largely intact, and the ligand wedged between domains I and III of the EGFR subunit, preventing any bending in this subunit's domain II. In contrast, a large gap opened in the dimer interface of the ligand-free EGFR–Her2 heterodimer (this was consistently observed in two independent simulations) between the N-terminal portions of the domain IIs (*Figure 4B*; *Video 1*). This occurred as domain II of the EGFR subunit bent away from the dimer interface upon the removal of its bound ligand. Similarly, a gap at the dimer interface was also observed in two independent simulations of the Her2 homodimer (*Figure 4C*). Plots of the buried surface area show that the size of the gap fluctuated in the simulations (*Figure 4B*), but on average, was significantly lower in the ligand-free EGFR–Her2 heterodimer and Her2 homodimer than in the ligand-bound heterodimer.

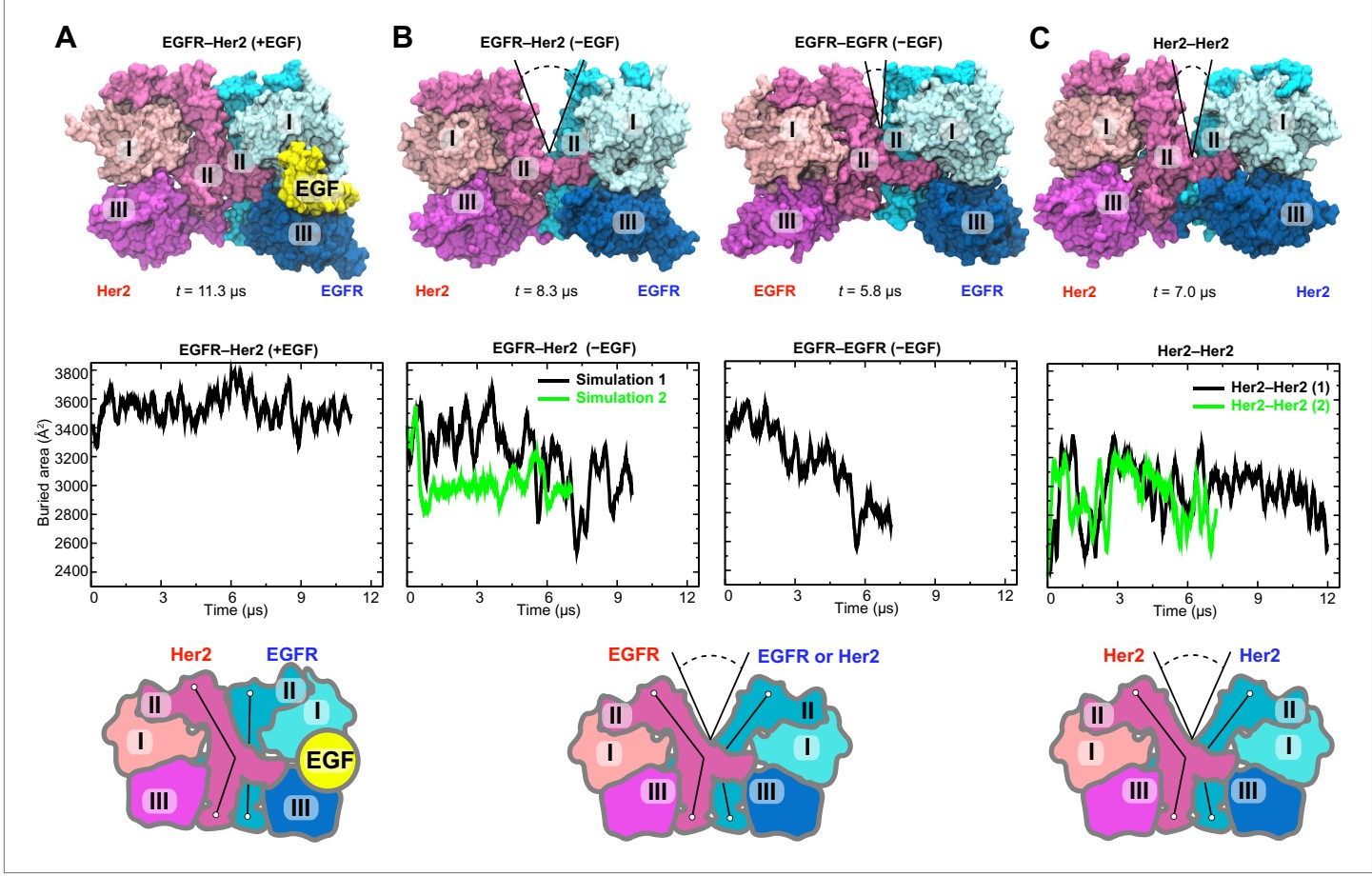

**Figure 4**. Simulations of the EGFR–Her2 heterodimer and the Her2 homodimer. (**A**) The EGFR–Her2 heterodimer with the ligand bound to EGFR ('+EGF'). (**B**) The ligand-free EGFR–Her2 heterodimer or EGFR–EGFR homodimer ('−EGF'). (**C**) The Her2 homodimer. Top: snapshots from the simulations. Middle: plots of the surface area buried within the dimer interface (counting the contributions from domains I, II, and III) as a function of simulation time (for EGFR–Her2 [−EGF] and Her2–Her2, results of two independent simulations are shown). Bottom: schematics illustrating the conformation of domain II and the dimer interface. Conformations of the bent and straight domain II are highlighted by the black lines, and the gap in the dimer interface is indicated by a V-shaped outline when present. For clarity, domain IV is not shown.

In terms of their location and potential impact on dimer stability, the gaps observed in our simulations are reminiscent of the one in the crystal structure of ligand-free dEGFR dimer (*Alvarado et al., 2009*, *2010*). The dEGFR gap results in a reduced dimer interface area of ~2300 Å², compared to ~3400 Å² in the 1- and 2-ligand dEGFR dimers, where the gap is closed (*Figure 2*). This reduction of the dimer interface area is believed to account for the relatively weak dimerization affinity of the ligand-free dEGFR, which is ~30 times lower than that of ligand-bound dEGFR. *Figure 4* shows that the reduction of the dimer interface is similarly observed in our simulations of the ligand-free EGFR–Her2 heterodimer and Her2 homodimer: when the gap is wide open, the surface area buried within the interface is ~2600 Å², compared to ~3500 Å² in the ligand-bound heterodimer. We thus suggest that the same structural mechanism that weakens ligand-free dEGFR homodimers (*Alvarado et al., 2009*, *2010*) also disfavors ligand-free EGFR–Her2 dimerization and Her2 homodimerization. On the other hand, ligand binding in the EGFR subunit helps to keep domain II straight, which prevents the opening of the gap and stabilizes the dimer interface of EGFR–Her2 heterodimer (*Figure 4*). This is reminiscent of the recent finding (*Liu et al., 2012*) that one bound ligand is sufficient to activate an EGFR dimer.

## Her3–Her2 heterodimer
Among the dimerization partners of Her2, Her3 is particularly notable in that the Her3–Her2 heterodimers are potent signaling units, and in that the normal and pathogenic signaling through Her2 and Her3

5.8us

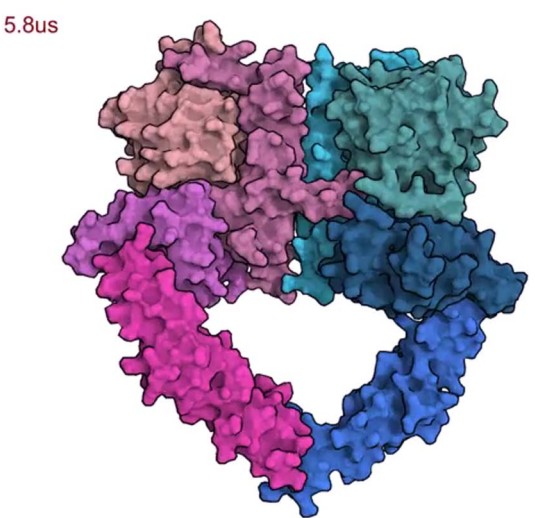

**Video 1**. The simulation of the ligand-free EGFR–Her2 ectodomain heterodimer with Her2 colored red and EGFR blue. A gap between the N-terminal portion of each subunit's domain II develops during the simulation. DOI: 10.7554/eLife.00708.007

relies strongly on Her3–Her2 dimerization (*Baselga and Swain, 2009*; *Lemmon, 2009*). We thus studied the Her3–Her2 heterodimer in ways similar to our study of EGFR–Her2 heterodimers and the Her2 homodimer. We first modeled the Her3–Her2 heterodimer using the resolved crystal structure of the Her3 monomer (*Cho and Leahy, 2002*) and our model of the EGFR–Her2 heterodimer as templates ('Materials and methods'). The EGF-like domain of the Her3 ligand heregulin-alpha (HRG) was then introduced into the Her3 binding site between domains I and III, in accordance with the pose of EGFR-bound EGF.

The resulting model of the Her3–Her2 heterodimer was then simulated, first with HRG bound, in which case the dimer interface remained stable (*Figure 5A*). After ~4 μs of simulation the ligand was removed, and the resulting ligand-free heterodimer was then simulated again. Without the ligand, the Her3 subunit underwent conformational changes similar to those described above for the ligand-free EGFR subunit, including a bending of domain II. As a result, a gap again emerged in the

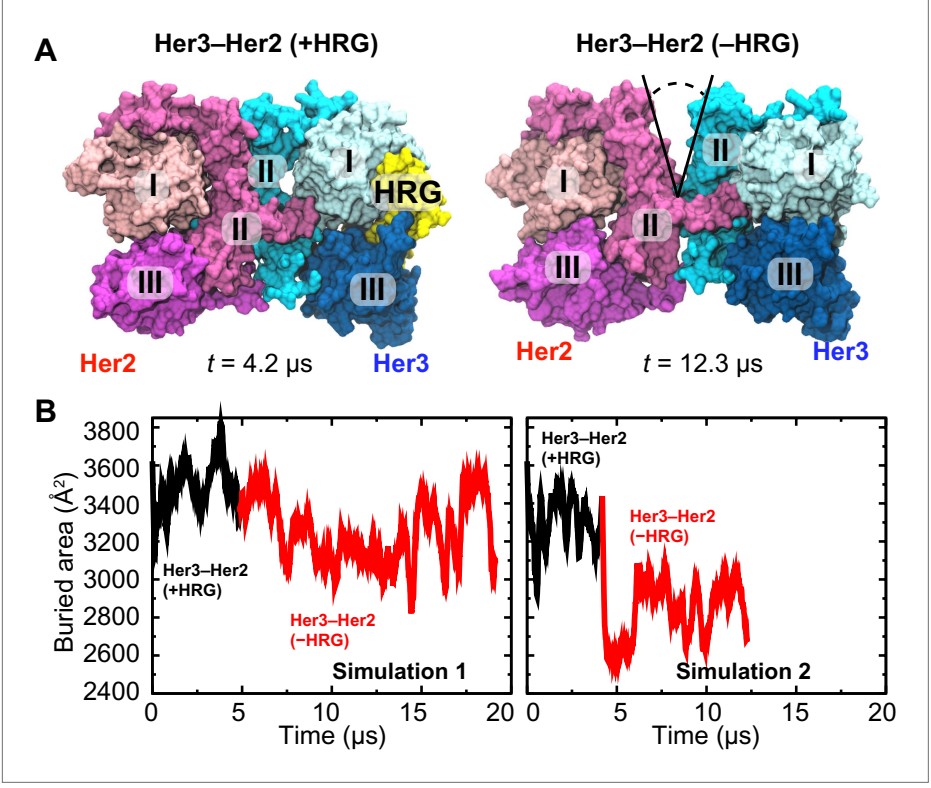

**Figure 5**. Her3–Her2 heterodimer. (**A**) Snapshots from the simulations of the Her3–Her2 heterodimer with (left) and without (right) HRG bound to Her3. At the end of the simulation with HRG bound to Her3, HRG was removed, and the resulting system was resolvated and further simulated without the ligand. A gap opened in the dimer interface, as illustrated by the snapshot on the right. For clarity, these images omit domain IV. (**B**) The surface area buried within the dimer interface, counting the contributions only from domains I, II, and III, plotted as a function of time. Two independent sets of two simulations each (with and without HRG) are shown.

dimer interface. In the first series of simulations the gap in the Her3–Her2 heterodimer was often partially closed, but re-opened repeatedly and resulted in a significant decrease in the buried surface area in the dimer interface. In the second series, the gap was steadily open once the ligand was removed (*Figure 5B*). These observations suggest that Her3–Her2 heterodimerization, similar to that of EGFR–Her2, is promoted by the stabilization of the dimer interface following ligand binding.

## Atomic detail of the dimer interfaces in the asymmetric dimers

Our simulations suggest that ligand binding promotes the formation of asymmetric dEGFR or EGFR-family dimers by allowing specific favorable interactions between the domain II N-terminal regions of the two subunits. We found that the asymmetric dimers we examined—the EGFR–Her2 and Her3–Her2 heterodimers, the 1-ligand EGFR homodimer, and the 1- and 2-ligand dEGFR homodimers—all shared certain key atomic-level interactions between subunits. In particular, hydrogen bonds between Gln194 (in EGFR or Her3 numbering) of the ligand-bound subunit and Cys213 and His215 (in Her2 numbering; *Figure 6A*) of the ligand-free subunit (*Figure 6B*) were particularly stable. The presence of these hydrogen bonds is consistent with the reported crystal structures of the dEGFR asymmetric dimers after a 180° $\chi_2$ rotamer flip at the histidine (which would presumably be consistent with the same electron density). In contrast, these hydrogen bonds are not present in the crystal structure or in simulations of the symmetric 2-ligand EGFR dimer, and they are precluded by the gap in the dimer interface of all ligand-free dimers we examined (including the ligand-free dEGFR crystal structure and our simulations of ligand-free EGFR and Her2 homodimers and EGFR–Her2 heterodimers). Our observations suggest an important role for Gln194 of the ligand-bound subunit in stabilizing 1-ligand dimers. Notably, the glutamine residue is conserved in EGFR, Her3, Her4, and dEGFR, which bind ligands, but not in Her2 (*Figure 6A*), which does not. Mutating Gln194 in EGFR and Her3 may thus hinder their heterodimerization with Her2, and mutating the corresponding residue (Gln189) in dEGFR may hinder its homodimerization.

Certain inter-subunit interactions appeared to be unique to Her2 heterodimers. In particular, Arg228 of EGFR interacted stably with Glu243 of Her2 through a salt bridge in our simulations of the ligand-bound EGFR–Her2 heterodimer (*Figure 6B*). Similarly, Arg228 of Her3 interacted stably with the same glutamate through a water bridge in our simulations of the Her3–Her2 heterodimer. Although this arginine is present in both EGFR and Her3, it is not present in Her2, Her4, or dEGFR. Notably, EGFR and Her3 are the two common dimerization partners of Her2 in the EGFR family, and only Her2 bears a negatively charged residue at the position of the Glu243 (*Figure 6A*). This is consistent with the notion that Her2 has been fine-tuned for heterodimerization with EGFR and Her3.

Experimentally, heterodimerization between Her2 and ligand-bound EGFR or Her3 ectodomains is much weaker than homodimerization of EGFR, Her4, or dEGFR ectodomains (*Horan et al., 1995*; *Ferguson et al., 2000*). Our simulations suggest that this difference in stability may be due to unique conformational dynamics in the dimerization arm of Her2. The dimerization arm plays a key role in maintaining dimer stability, and dimerization arm-mediated interactions are very similar in crystal structures of dEGFR and human EGFR dimers (*Alvarado et al., 2010*). In our simulations, however, the dimerization arm of Her2 in the EGFR–Her2 dimer deviated significantly from its initial conformation in the course of the simulation, whereas the dimerization arms of EGFR and dEGFR in homodimers did not (*Figure 6C*). A cation–π interaction at the tip of the dimerization arm in the 1-ligand EGFR dimer between Tyr251 and Arg285 of the unliganded subunit (*Figure 6C*) appears to be particularly important in stabilizing the dimer. The residue at the position of Tyr251 is conserved as tyrosine or phenylalanine in dEGFR and all members of the human EGFR family. The residue at the position of Arg285 is conserved as arginine, except in Her2, where it is replaced by leucine (Leu291; *Figure 6A*). It is tempting to suggest that this missing arginine in Her2 may explain its weak ectodomain heterodimerization compared to the homodimerization of dEGFR, EGFR, and Her4, and that an L291R mutation in Her2 may promote Her2 heterodimerization and facilitate the crystallization of a Her3–Her2 or EGFR–Her2 dimer.

## Discussion

Given the implication of Her2 in disease and its central role in mediating cell signaling, elucidation of the molecular mechanisms regulating Her2 activation is of great importance. The long-timescale MD simulations reported here support the notion of a simple and evolutionarily conserved structural mechanism controls homo- and heterodimerization of Her2, and thereby Her2 activation. Our simulations revealed inherent instability in the dimer interfaces of the ligand-free EGFR–Her2 and Her3–Her2

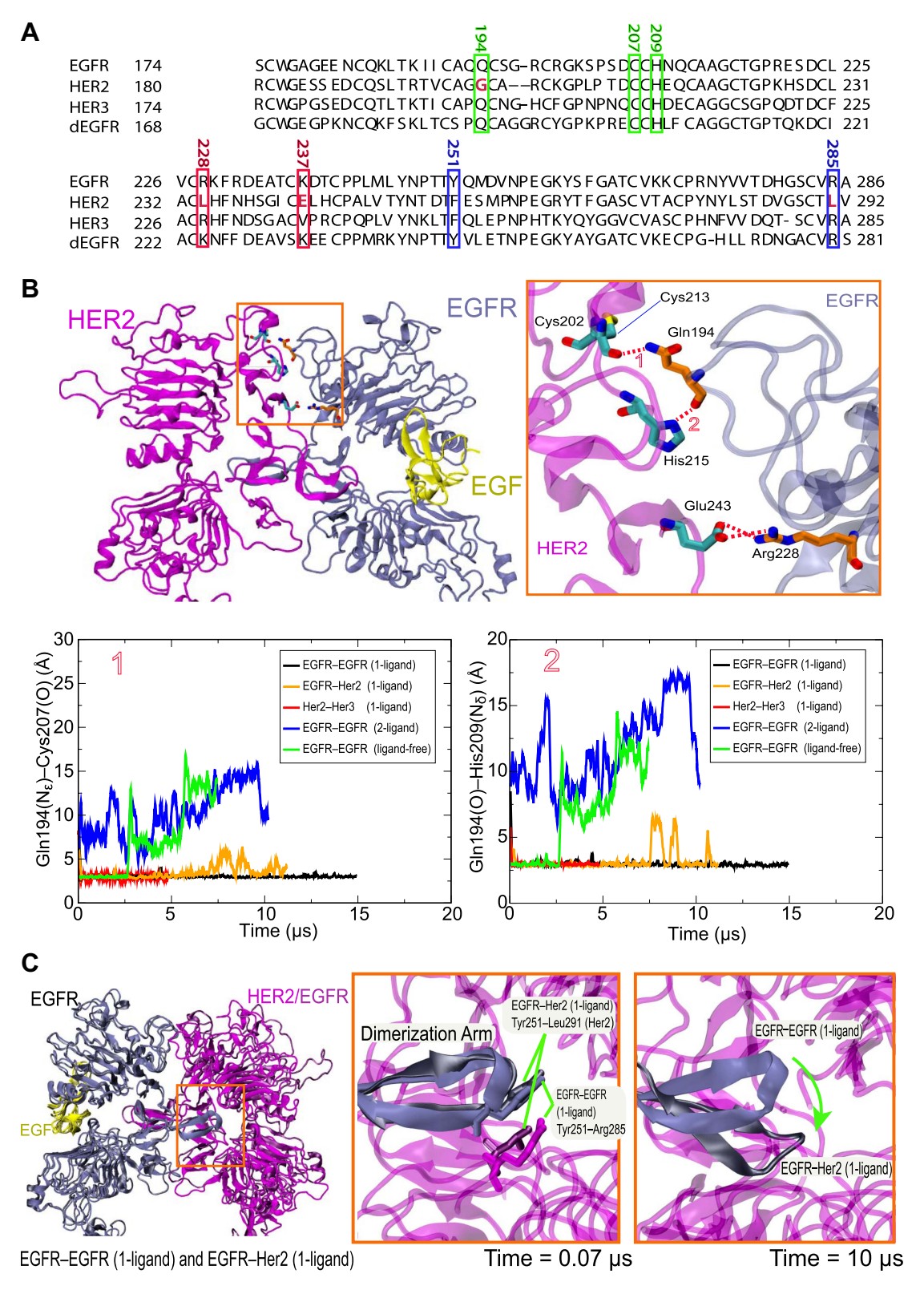

**Figure 6**. Details of the dimer interfaces in Her2 heterodimers. (**A**) The sequence alignments of EGFR, Her2, Her3, and dEGFR for the part of domain II corresponding to EGFR residues 167–286. Three sets of residues (green, red, and blue, respectively), which are involved in three sets of important interactions at the dimer interface, are highlighted. (**B**) Top: the role of EGFR residue Gln194 in the EGFR–Her2 dimer interface. Hydrogen bonds and
*Figure 6. Continued on next page*

*Figure 6. Continued*

salt bridges are indicated by red dashed lines. Bottom: the distances between the key atoms involved in these hydrogen bonds are shown as functions of time. (**C**) Left: superimposition of 1-ligand EGFR homodimer and 1-ligand EGFR–Her2 heterodimer, with dimerization arms highlighted by the red box. Middle and right: Her2's dimerization arm in the EGFR–Her2 heterodimer is conformationally less stable than the dimerization arm of the ligand-free subunit of the 1-ligand EGFR homodimer, likely because the Tyr251–Arg285 cation–π interaction is only present in the latter dimer. Dark blue is used for the EGFR dimerization arm in the EGFR homodimer, and dark gray for the EGFR dimerization arm in the EGFR–Her2 heterodimer.

heterodimers and the Her2 homodimer, and indicated that the binding of a single ligand is sufficient to stabilize the heterodimer interfaces. These observations complement the crystallographic analysis, shedding further light on the structural mechanism underlying the preference of Her2 to partner with a ligand-bound EGFR or Her3, rather than with Her2 or ligand-free EGFR or Her3 (*Lemmon, 2009*; *Graus-Porta et al., 1997*; *Nagy, et al., 2010*).

Our simulations showed the conformation of domain II, which is determined by ligand binding, to be critical to the stability of the dimer interface. A ligand bound to EGFR (or to another member of the family) acts as a wedge that pushes domains I and III apart, straightening the otherwise bent domain II (*Alvarado et al., 2009*). (Domain II is bent in the crystal structures of ligand-free EGFR, Her2, Her3, Her4, and dEGFR.) We simulated the transition from the straight to the bent conformation of domain II upon removal of the bound ligand in EGFR and Her3 and the consequent development of the gap in the dimer interface. Our simulations suggest that, for the gap to open, both domain IIs of the two subunits in a dimer need to adopt a bent conformation.

One feature unique to the dimerization and activation of receptors of the EGFR family (Her2 included), but not to other families of receptor tyrosine kinases (*Lemmon and Schlessinger, 2010*), is that the dimer interface is mediated by the receptors and does not involve any direct contribution from the bound ligands. Instead of directly participating in the dimerization, a bound ligand regulates the conformation of domain II and potentiates it for dimerization. In this context, it is not surprising that the seemingly minor effect of ligand binding on the conformation of domain II in the heterodimerization partners of Her2 demonstrated by our simulations is of great importance to the regulation of Her2 activity. Our consistent observation of the gap that disrupts the dimer interfaces in the ligand-free EGFR–Her2 and Her3–Her2 heterodimers, and in Her2 and dEGFR homodimers, as well as the fact that domain II is bent in crystal structures of ligand-free EGFR, Her2, Her3, Her4, and dEGFR, suggests that the structural mechanism for the control of dimerization has been preserved from *Drosophila* to the human EGFR family.

## Materials and methods

### Modeling of dimers

Because EGFR, Her2, and Her3 are closely related phylogenetically and share a high degree of sequence identity, we used the EGFR ectodomain homodimer as a structural template for the modeling of the Her2 ectodomain homodimer and the EGFR–Her2 and Her3–Her2 ectodomain heterodimers. The sequence identity between the ectodomains of EGFR and Her2 is 40%, while that between the ecto-domains of EGFR and Her3 is 41%. In the structural alignments for the modeling, we used the region corresponding to EGFR residues 240–309, since this is the part of domain II that appears to be essential for dimerization (*Dawson et al., 2005*), and since in our simulations of EGFR the N-terminal portion of domain II exhibited bending around the hinge situated approximately at residue 240 (*Figure 3*). In this region, sequence identity to EGFR is 50% and 46% for Her2 and Her3, respectively (*Figure 6A*).

The Her2 subunit of the EGFR–Her2 heterodimer was modeled based on the Her2 monomer crystal structure from PDB entry 3BE1 (*Bostrom et al., 2009*), and missing residues in domains I–III were filled from PDB entries 1S78 (*Franklin et al., 2004*) and 2A91 (*Garrett et al., 2003*). The Her2 monomer was aligned with the ligand-free EGFR subunit from the 1-ligand EGFR dimer using EGFR residues 240–309 (and the corresponding residues of Her2) for reference. The part of domain IV of Her2 that is missing in the crystal structures (beyond residue 608) was modeled based on the EGFR structure. The minor clashes between Her2 and EGFR in the original model were resolved by adjusting the side chains of the amino acids involved. To obtain the Her2 homodimer, the Her2 monomer was used to replace the EGFR subunit in the ligand-free EGFR–Her2 heterodimer.

The Her3–Her2 heterodimer was modeled based on our model of the EGFR–Her2 dimer taken at $t = 0$ (i.e., before simulation). Individual domains from the crystal structure of the tethered Her3

ectodomain (PDB entry 1M6B [*Cho and Leahy, 2002*]) were aligned on the respective domains of the EGFR template. Because domain II is straight in the EGFR template and bent in the Her3 crystal structure, the alignment was carried out in three separate steps. Specifically, Her3 residues 166–188, 189–238, and 239–307 were separately aligned with their EGFR counterparts. The residues at the borders between the parts of Her3 that were aligned as rigid bodies were then adjusted to ensure appropriate connectivity; a few minor clashes between the domains of Her3 and between Her3 and Her2 were eliminated by adjusting amino acid side chains. HRG was taken from PDB entry 1HAE (*Jacobsen et al., 1996*) and aligned with the EGF bound to the EGFR template, using EGF's residues 26–46 for reference (as this part appears to be most similar structurally between the two).

Domains IVs of receptors of the EGFR family are generally more flexible than the other ectodomains and are commonly unresolved in crystal structures. Thus, structural analysis of domain IVs starting from homology models may be particularly challenging. Although domain IVs were included in our models and simulations, they were not a main concern of this study.

## Simulation details

The simulations were performed on the special-purpose supercomputer, Anton (*Shaw et al., 2009*), using the CHARMM22* force field (*MacKerell et al., 1998*; *Piana et al., 2011*) for proteins and TIP3P (*Jorgensen et al., 1983*) as the water model. The simulated systems were solvated in water with NaCl ($Na^+$ ions were first added to neutralize the net charge of the system, and then equal numbers of $Na^+$ and $Cl^-$ were added so that the concentration of $Na^+$ reached 0.15 M), with residues set to their dominant protonation states at pH 7. As an equilibration stage, the protein backbone atoms were first restrained to their initial positions, using a harmonic potential with a force constant of 1 kcal mol$^{-1}$ Å$^{-2}$. The force constant was linearly scaled down to zero over 50 ns. Simulations were performed in the NPT ensemble for the equilibration step and in the NVT ensemble afterwards, with T = 310 K, p=1 bar, and Berendsen's coupling scheme (*Berendsen et al., 1984*) with one temperature group. Water molecules and all bond lengths to hydrogen atoms were constrained using M-SHAKE (*Kräutler et al., 2001*). Van der Waals and short-range electrostatic interactions were cut off at 12.5 Å. Long-range electrostatic interactions were calculated using the *k*-space Gaussian split Ewald method (*Shan et al., 2005*) with a 64 × 64 × 64 mesh. The simulation time step was 1 fs for the equilibration stage and 2.5 fs for production simulations; the r-RESPA integration method was used with long-range electrostatics evaluated every 5 fs (*Tuckerman et al., 1992*).

The simulated systems included the EGFR ectodomain dimers with two EGF ligands (three simulations) or one ligand (also three simulations); 1-ligand EGFR–Her2 ectodomain heterodimer (one simulation), ligand-free heterodimer (two simulations), Her2 homodimer (two simulations), 1-ligand Her3–Her2 heterodimer (two simulations), and ligand-free Her3–Her2 heterodimer (two simulations); and ligand-free dEGFR dimer (two simulations starting from the 2-ligand crystal structure and one from the ligand-free crystal structure). The size of the simulated systems was ~270,000 atoms, in a periodic box of approximately 140 × 140 × 140 Å$^3$. Modeling, analysis, and visualization were performed using VMD (*Humphrey et al., 1996*). The data for buried area shown in the figures was averaged over a 200-ns window.

Repeated simulations of a given system produced largely similar results. In both simulations of the ligand-free EGFR–Her2 dimer, for example, a gap consistently developed at the dimer interface (*Figure 4B*). Three simulations of the 1-ligand EGFR homodimer (*Figure 3D*), generated average structures with a root-mean-squared deviation of 1.9–4.8 Å from one another, as measured based on the Cα atoms of domains I–III. By the same measurement, the two crystal structures of the 2-ligand EGFR homodimer (PDB entries 1IVO and 1MOX) differ from one other by 4.0 Å.

## Acknowledgements

We thank John Kuriyan, Michael Eastwood, and Stefano Piana for helpful discussions, Ansgar Philippsen for assistance with graphics, and Mollie Kirk and Berkman Frank for editorial assistance. We also thank the reviewers for their helpful suggestions.

## Additional information

### Funding

The authors declare that there was no external funding for this work.

## Author contributions

AA, Conception and design, Preparing and performing simulations, Analysis and interpretation of data, Drafting or revising the article; YS, Conception and design, Analysis and interpretation of data, Drafting or revising the article; ETK, Preparing and performing simulations; ROD, Interpretation of data, Drafting or revising the article; DES, Conception and design, Interpretation of data, Drafting or revising the article

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
