## [Decision Letter]

Thank you for sending your work entitled “Her2 activation mechanism reflects evolutionary preservation of asymmetric ectodomain dimers in the human EGFR family” for consideration at *eLife*. Your article has been favorably evaluated by a Senior Editor and 3 reviewers, one of whom, Axel Brunger, is a member of our Board of Reviewing Editors.

The Reviewing editor and the other reviewers discussed their comments before we reached this decision, and the Reviewing editor has assembled the following comments to help you prepare a revised submission.

General assessment:

The mechanism of Her2 activation and signaling is of considerable importance considering its role in human disease (including cancer) and as a drug target. The available biochemical data suggest that the activation process upon ligand binding requires formation of a heterodimer consisting of Her2 and Her3. Her2 lacks ligand binding (or binding is weak), whereas Her3 does not have significant kinase activity. Thus, it is only upon hetero-dimerization that ligand binding can initiate kinase activity. Yet, structural studies by x-ray crystallography of the human EGFR-family receptors ectodomains showed only symmetric dimers with ligands bound in both monomers. Intriguingly, crystal structures of *Drosophila* EGFR ectodomains showed asymmetric dimers with only one ligand bound. The question, however, was if the *Drosophila* crystal structures are representative, what is going on for the human EGFR receptors as well. This work now suggests that this is indeed the case: starting from the crystal structures of the human Her2 homodimer as well as the Her2-Her3 heterodimer, long-timescale (microsecond) molecular dynamics simulations were performed. The simulations were first successfully tested starting from the *Drosophila* heterodimer structures.

Highlight:

The simulations show that a model of the human Her2 homodimer is not stable, but, rather, that the dimer interface opens to produce a large gap. In contrast, a model of the Her2-Her3 heterodimer with one bound ligand was stable. This work thus suggests that the asymmetric *Drosophila* EGFR ectodomain crystal structures indeed represent the norm, rather than the exception.

While this is an important finding, the simulations should allow the authors to draw some mechanistic conclusions about how the conformational changes within the various dimers occur and how the different interfaces are stabilized. Such additional mechanistic insights would also illustrate the power of their long-time simulations and considerably strengthen this work.

Required revisions:

1) By taking the richness of the MD simulations and reducing them to cartoons in Figures 3, 4 and 5, the authors do not go much beyond simply confirming conclusions already drawn from the studies of *Drosophila* EGFR. Surely the purpose of simulations such as those described here is to generate informative models that make experimentally testable predictions? In the crystallographic studies of dEGFR, identification of residues involved in interactions across the asymmetric dimer interface was informative. The nature of this new interface is crucial for understanding the fly receptor, but here the authors unfortunately give no details for the analogous interfaces in the human heterodimers studied. This omission deprives those in this field of the most useful information to be gleaned from the simulations presented. Are the interfaces (e.g., at the top of Figure 4) similar to those seen in dEGFR? Do they involve conserved residues?

2) The “structural explanation for the preference of Her2 to partner with a ligand-bound EGFR or Her3, rather than with Her2 or ligand-free EGFR or Her3” is not really a result of this paper. Quite frankly, this was already clear from the *Drosophila* work. What the present (excellent) study does add to this, though, is the potential of a detailed view – yet the authors disappointingly describe no details that can be used for designing experiments. The brief discussion does not suffice and key areas of contact are omitted from Figure 6. Are there key residues in the domain II interface in the asymmetric hEGFR dimer (outside the region shown in Figure 6) that are well conserved? Can predictions be made for mutations that might selectively destabilize the asymmetric hEGFR dimer but not the symmetric 2-ligand dimer? These would be useful outcomes of the modeling studies and analysis of these questions are likely to yield valuable insight in and of itself.

3) Why are heterodimers formed between the EGFR and Her2 (or Her2 and Her3) ectodomains so much weaker than EGFR-EGFR ectodomain homodimers? This is a clear experimental result in the literature. In fact, EGFR-Her2 ectodomain heterodimers have not yet been directly observed, following ligand binding, although there are hints from biophysical studies that weak Her2-Her3 ectdomain dimers may form (this varies from study to study). By contrast, EGFR and dEGFR ectodomain dimers are strong (at least 50-fold stronger). The simulations might address the question of different dimerization strengths. By ignoring the well-established affinity differences – when taken at face value – the presentation in this work is a little misleading. For example, what do the authors mean by suggesting that the Her3-Her2 heterodimers are particularly robust? If they mean that the affinity of Her3 for Her2 is particularly strong, they are not correct – there is no evidence for this. Evidence for heterodimer formation is qualitative at best, and the only quantitative data show that EGFR-EGFR homodimers and Her4-Her4 homodimers are much “tighter” than any homodimer, at least where ectodomains are concerned. Analysis and description of the dimer interfaces could provide insight into this issue too.

4) Although this manuscript is well-written and beautifully illustrated, it is spoiled by its logical framework. For example, Figure 1 shows the symmetrical human EGFR (hEGFR) dimer. It also shows tethered and untethered forms of the monomer, but specifically does not address them. If these are not addressed, why illustrate them. The fact that Her2 without a ligand looks superficially like hEGFR with a ligand makes one want to assume that this is how Her2 works. Also, it should be made clearer (e.g., in the figure caption) which specific crystal structures of which domains and homo or heterodimers have been determined, along with their PDB IDs.

5) Figure 2 shows the symmetric *Drosophila* EGFR (dEGFR) dimer, the logical counter-part of the symmetric human EGFR dimer in Figure 1. It also shows the effect of a simulation on dEGFR without both its ligands. Figure 3 shows a simulation of hEGFR without one its ligands. It seems that both the human and *Drosophila* x-ray structures should be introduced together so as to emphasize their different symmetries.

6) The results of parallel simulations in both hEGFR and dEGFR need to be shown. This may require additional work as hEGFR has been simulated without one ligand whereas dEGFR has been simulated without both ligands.

7) The results of the MD simulations seem reproducible in that two independent several-micro-second simulations give similar results for this huge system with almost 300,000 atoms. This remarkable result could be emphasized more explicitly by comparison of the final structures of the respective pairs of simulations.

---

## [Author Response]

*The simulations show that a model of the human Her2 homodimer is not stable, but, rather, that the dimer interface opens to produce a large gap. In contrast, a model of the Her2-Her3 heterodimer with one bound ligand was stable. This work thus suggests that the asymmetric Drosophila EGFR ectodomain crystal structures indeed represent the norm, rather than the exception*.

*While this is an important finding, the simulations should allow the authors to draw some mechanistic conclusions about how the conformational changes within the various dimers occur and how the different interfaces are stabilized. Such additional mechanistic insights would also illustrate the power of their long-time simulations and considerably strengthen this work*.

We thank the reviewers for these comments. In the revised manuscript, we have dedicated an updated figure (Figure 6) and a new sub-section of the Results to highlight the residue–residue interactions that are key to the stability of these asymmetric EGFR family dimers. We have also added suggestions for mutations that may enhance the stability of these asymmetric dimers and potentially facilitate their crystallization.

*Required revisions*:

*1) By taking the richness of the MD simulations and reducing them to cartoons in Figures 3, 4 and 5, the authors do not go much beyond simply confirming conclusions already drawn from the studies of* Drosophila *EGFR. Surely the purpose of simulations such as those described here is to generate informative models that make experimentally testable predictions? In the crystallographic studies of dEGFR, identification of residues involved in interactions across the asymmetric dimer interface was informative. The nature of this new interface is crucial for understanding the fly receptor, but here the authors unfortunately give no details for the analogous interfaces in the human heterodimers studied. This omission deprives those in this field of the most useful information to be gleaned from the simulations presented. Are the interfaces (e.g., at the top of Figure 4) similar to those seen in dEGFR? Do they involve conserved residues*?

In the revised manuscript, we provide further details of the dimer interfaces observed in our MD simulations. In particular, we report residue-specific details of the interfaces of asymmetric EGFR family dimers in the new sub-section of the Results and the new version of Figure 6. Based on the conformational dynamics of the dimer interfaces, we determined that the interactions of Gln194 (EGFR numbering) with Cys213 and His215 (Her2 numbering), which are stable in our simulations of the 1-ligand EGFR homodimer and the Her2 heterodimers with ligand-bound EGFR or Her3, are critical to the stability of these asymmetric dimers. These interactions are also formed in the crystal structures of the asymmetric dEGFR dimers. We have added a discussion of these interactions, and of the conservation of the residues involved, to the revised manuscript.

*2) The “structural explanation for the preference of Her2 to partner with a ligand-bound EGFR or Her3, rather than with Her2 or ligand-free EGFR or Her3” is not really a result of this paper. Quite frankly, this was already clear from the* Drosophila *work. What the present (excellent) study does add to this, though, is the potential of a detailed view – yet the authors disappointingly describe no details that can be used for designing experiments. The brief discussion does not suffice and key areas of contact are omitted from Figure 6. Are there key residues in the domain II interface in the asymmetric hEGFR dimer (outside the region shown in Figure 6) that are well conserved? Can predictions be made for mutations that might selectively destabilize the asymmetric hEGFR dimer but not the symmetric 2-ligand dimer? These would be useful outcomes of the modeling studies and analysis of these questions are likely to yield valuable insight in and of itself*.

We agree that the overall structural mechanism of Her2 dimerization may largely be inferred from the structures of the dEGFR dimers, and that much of the value of our work lies in its potential to provide a detailed view of the relevant dimer interfaces. As the reviewers suggested, we have thus revised the manuscript to place more emphasis on the atomic details of our Her2 heterodimer models, with a particular focus on the dimer interfaces. We have also revised the relevant high-level statements in the manuscript to reflect this understanding. The new version of Figure 6 presents important details of the interactions between the N-terminal regions of the domain IIs in the asymmetric dimers, in addition to the interactions around the “dimerization arms”. Because the interactions centered on Gln194 (EGFR numbering) are stable in the asymmetric EGFR family dimers but not in their symmetric counterparts, we suggest in the revised manuscript that mutation of this residue would destabilize the former more than the latter dimers. Additionally, we have now highlighted the salt bridge between Arg228 (EGFR numbering) and Glu243 (Her2 numbering), which is likely important to the stability of the Her2 dimers. In the revised manuscript we further suggest that mutating Arg228 may hinder the heterodimerization of EGFR and Her3 with Her2, but not the homodimerization.

*3) Why are heterodimers formed between the EGFR and Her2 (or Her2 and Her3) ectodomains so much weaker than EGFR-EGFR ectodomain homodimers? This is a clear experimental result in the literature. In fact, EGFR-Her2 ectodomain heterodimers have not yet been directly observed, following ligand binding, although there are hints from biophysical studies that weak Her2-Her3 ectdomain dimers may form (this varies from study to study). By contrast, EGFR and dEGFR ectodomain dimers are strong (at least 50-fold stronger). The simulations might address the question of different dimerization strengths. By ignoring the well-established affinity differences – when taken at face value – the presentation in this work is a little misleading. For example, what do the authors mean by suggesting that the Her3-Her2 heterodimers are particularly robust? If they mean that the affinity of Her3 for Her2 is particularly strong, they are not correct – there is no evidence for this. Evidence for heterodimer formation is qualitative at best, and the only quantitative data show that EGFR-EGFR homodimers and Her4-Her4 homodimers are much “tighter” than any homodimer, at least where ectodomains are concerned. Analysis and description of the dimer interfaces could provide insight into this issue too*.

The reviewers are correct that there is no evidence indicating that the affinity of Her3 for Her2 is particularly strong. We have revised the sentence of the original manuscript that appeared to suggest otherwise, and we have clarified the point we intended to convey, which was the functional importance of full-length Her3–Her2 heterodimers.

In the revised manuscript, we have also attempted to address the reviewers’ question of why ectodomain heterodimerization of Her2 and Her3 is weaker than ectodomain homodimerization of dEGFR, EGFR, and Her4. To elucidate the structural basis for the different dimerization strengths, we re-examined the interactions at the dimerization interfaces. In the revised manuscript, we now identify variance at key residues that may explain the strength of dEGFR and human EGFR homodimers relative to Her2 heterodimers. Specifically, we suggest that a cation–π interaction between Tyr251 and Arg285 (EGFR numbering) in the homodimers may be key to the stabilization of the dimerization arms in the dimer interfaces. Furthermore, we suggest that the replacement of the arginine in Her2 by a leucine, which distinguishes it from dEGFR and the other members of the EGFR family, may explain the weak ectodomain heterodimerization of Her2 ectodomains relative to the homodimerization of dEGFR and human EGFR ectodomains. We further speculate that the L291R Her2 mutation, which may stabilize Her2 heterodimers with EGFR and Her3, may facilitate crystallization of Her2–Her3 and Her2–EGFR dimers.

*4) Although this manuscript is well-written and beautifully illustrated, it is spoiled by its logical framework. For example, Figure 1 shows the symmetrical human EGFR (hEGFR) dimer. It also shows tethered and untethered forms of the monomer, but specifically does not address them. If these are not addressed, why illustrate them. The fact that Her2 without a ligand looks superficially like hEGFR with a ligand make one want to assume that this is how Her2 works. Also, it should be made clearer (e.g., in the figure caption) which specific crystal structures of which domains and homo or heterodimers have been determined, along with their PDB IDs*.

As suggested, we have removed the tethered conformation from the figures and specified which crystal structures the cartoon diagrams represent by including their PDB IDs. We have also attempted to improve the manuscript’s logical framework by presenting the dEGFR dimer together with the symmetric EGFR dimer in the revised Figure 1, as described below in our response to reviewer comment 5.

*5) Figure 2 shows the symmetric* Drosophila *EGFR (dEGFR) dimer, the logical counter-part of the symmetric human EGFR dimer in Figure 1. It also shows the effect of a simulation on dEGFR without both its ligands. Figure 3 shows a simulation of hEGFR without one its ligands. It seems that both the human and* Drosophila *x-ray structures should be introduced together so as to emphasize their different symmetries*.

In the revised manuscript, the 2-ligand asymmetric dEGFR dimer is now juxtaposed with the symmetric EGFR dimer in Figure 1 to emphasize their different symmetries. We have also revised the text so that the two dimers are introduced together.

*6) The results of parallel simulations in both hEGFR and dEGFR need to be shown. This may require additional work as hEGFR has been simulated without one ligand whereas dEGFR has been simulated without both ligands*.

In the revised manuscript we now report parallel simulations of the ligand-free EGFR dimer and show that a gap develops at the dimer interface, as we had previously observed for the ligand-free dEGFR dimer (Figure 4).

*7) The results of the MD simulations seem reproducible in that two independent several- micro-second simulations give similar results for this huge system with almost 300,000 atoms. This remarkable result could be emphasized more explicitly by comparison of the final structures of the respective pairs of simulations*.

We agree that the reproducibility of our simulations warrants a more explicit discussion. At the end of the Methods section in the revised manuscript, we now report that the average structures observed in three simulations of the 1-ligand EGFR homodimer differ by 1.9–4.8 Å in terms of the RMSD of the Cα atoms, which is comparable to the Cα RMSD between the two available crystal structures of the 2-ligand EGFR dimer (4.0 Å). This comparison differs only slightly from the one the reviewers suggested in that the average, rather than the final, conformation of each simulation is used, in light of the fact that the average conformation of a simulation generally represents the ensemble of conformations sampled by the simulation better than the final one.